# Plasmon-Mediated Oxidation Reaction on Au/p-Cu₂O: The Origin of Hot Holes

**Philipp Hawe [1,†], Vitor R. R. Silveira [1,†], Robert Bericat Vadell [1,†], Erik Lewin [1] and Jacinto Sá [1,2,*]**

1.   Department of Chemistry-Angstrom, Uppsala University, 751 20 Uppsala, Sweden;
Philipp.Hawe.5901@student.uu.se (P.H.); vitor.silveira@kemi.uu.se (V.R.R.S.);
robert.bericat.vadell@kemi.uu.se (R.B.V.); erik.lewin@kemi.uu.se (E.L.)
2.   Institute of Physical Chemistry, Polish Academy of Sciences (IChF-PAN), 01-224 Warsaw, Poland
*   Correspondence: jacinto.sa@kemi.uu.se; Tel.: +46-18-471-6806
†   Equal contribution.

**Abstract:** More sustainable solutions are needed to produce chemicals and fuels, mainly to face rising demands and mitigate climate change. Light, as a reagent, has emerged as a route to activate small molecules, e.g., $H_2O$, $CO_2$, $N_2$, and make complex chemicals in a process called photocatalysis. Several photosystems have been proposed, with plasmonic technology emerging as one the most promising technologies due to its high optical absorption and hot-carrier formation. However, the lifetime of hot carriers is unsuitable for direct use; therefore, they are normally coupled with suitable charge-accepting materials, such as semiconductors. Herein, a system is reported consisting of Au supported in p-Cu₂O. The combination of p-Cu₂O intrinsic photoactivity with the plasmonic properties of Au extended the system's optical absorption range, increasing photocatalytic efficiency. More importantly, the system enabled us to study the underlying processes responsible for hot-hole transfer to p-Cu₂O. Based on photocatalytic studies, it was concluded that most of the holes involved in aniline photo-oxidation come from hot-carrier injections, not from the PIRET process.

**Keywords:** plasmon hybrid system; hole transfer mechanism; ultrafast transient spectroscopy; photocatalysis

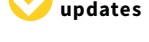



## 1. Introduction

The steadily rising consumption of energy and consumer products is becoming increasingly challenging. The resulting high demand for the respective production processes and the ensuing high rate of pollution and $CO_2$ emissions [1] calls for a more sustainable solution. Nature's photosynthesis inspires the development of novel chemical processes that utilise light as a reagent. Exploring photochemistry and photoredox catalysis can reduce dependence on fossil fuels, therefore placing humanity on a path towards a more sustainable industry. At present, energy generation from solar power is a developed and widely applied technique. A step forward from power generation is the photocatalytic production of chemicals and fuels, known as photoredox catalysis and artificial photosynthesis, respectively.

Since the early studies, where a $TiO_2$ semiconductor was utilised as a catalyst for light-driven water splitting, materials used in photo processes have increased in sophistication and performance. One of the significant issues with wide-bandgap semiconductor photocatalysis is its narrow solar-spectrum absorption. The addition of plasmonic nanoparticles to the photocatalyst matrix is a possible approach to surpass this inherent limitation, through their localized surface plasmon resonance (LSPR). Besides, the additional absorption band of the plasmon, its fundamental property to focus the incident light onto the nanoparticle, increases the strength of the interaction between light and matter, resulting in the possibility of better light-harvesting. As a result, a higher electron generation efficiency is given [2].

The LSPR is the collective charge oscillation of the electron gas in the metal nanoparticle (NP). Hence, the valence band electrons of the metal nanoparticle are influenced by the incident electric field of the light, $\vec{E}$ along the respective direction of polarization, $\vec{P}$. Fulfilling the conditions of size and permittivity, the electromagnetic wave of the incident light can induce the collective oscillation of the electron gas, which is maximal absorption dependent on particle size. Non-radiative excitation of electrons leading to charges is dominant in small particles. Those charges are not in thermal equilibrium with the lattice and have higher effective energy; therefore, they are referred to as hot carriers [3,4]. The formation of the hot carriers on Au plasmons occurs via several processes, namely electron-electron (e-e) scattering, electron-phonon (e-ph) scattering and Landau damping [5]. The latter dominates on small NPs and is based on the interaction of the collective oscillation with the natural boundaries of the NP. This interface-bound collision leads to a loss of coherence in the plasmon after the formation of hot carriers [6,7].

Once the LSPR is established, the relaxation process of the formally coherent oscillation leads to the generation of hot carriers and decays over time. This process is on the timescale of up to several nanoseconds [8]. Initially, the excitation with light of the wavelength at the LSPR frequency occurs. Showing the antenna effect of the plasmon at its resonance frequency, the generation of hot carriers occurs within the lifetime of the excited plasmon. During the excitation, the electrons of the system follow the Fermi–Dirac distribution according to the dominant thermal equilibrium in the lattice $T_{L0}$. This excited plasmon now decays in up to 100 fs. This indicates that the formally collective oscillation of the plasmon starts decaying and creates the separation of electron-hole pairs, which are highly non-thermalised and do not follow the Fermi–Dirac distribution. Those carriers start to decay through e-e collisions, which results in the thermal equilibrium following the Fermi–Dirac distribution, exhibiting charges at a higher effective temperature ($T_e \gg T_{L0}$). In the final stage, the created carriers undergo thermal relaxation through electron-phonon scattering, which causes thermal dissipation of the energy into the lattice of the noble metal and its surrounding environment.

To avoid energy dissipation via thermalisation and make use of the generated hot carriers, they can be injected into an adjacent semiconductor. This injection can be done by either the indirect (three-step) charge injection or by direct hot-carrier formation in the semiconductor [9]. In the indirect injection mechanism, hot carriers with sufficient energy to surpass the Schottky barrier at the metal-semiconductor heterojunctions are transferred ballistically from the plasmon to the semiconductor [10–13], while the others relax within the plasmonic structure. The direct injection (based on chemical interface damping) avoids the limitation of the Schottky barrier. Here, the excited plasmon relaxes and scatters directly to an interfacial state of the semiconductor, reducing the intrinsic loss by capturing the energy in the timescale of several femtoseconds. This allows the energy to be extracted from the plasmon before losses occur through thermal dissipation (e-e scattering) [14–16].

Either of the aforementioned pathways are interface-confined transfer processes, where either electrons or holes are transferred to the semiconductor. Plasmon-Induced Resonance Energy Transport (PIRET) relates to an energy-transfer mechanism [17] and results in both charges appearing at the semiconductor. The PIRET mechanism is similar to the fluorescence dependent, known as the process of Förster Resonance Energy Transfer (FRET). Like the FRET process, the PIRET process is of a non-radiative nature with the main difference being that the plasmonic resonance mediates the energy transfer [18]. A blueshift occurs in the overlap integrals of the donor (plasmon) and the acceptor (semiconductor). Hence, this mechanism results in an energy-upconverting energy transfer. Unlike the aforementioned charge-generation processes, this process does not inject or generate single carriers in the semiconductor. The transfer of energy into the adjacent semiconductor forms electron-hole pairs in the material through relaxation of the surface plasmons' resonance. This mechanistical difference, the coupling of the electromagnetic field with the semiconductor, rather than direct injection of carriers or tunnelling, explains the rather low-distance

dependence. This leads to an energy transfer that can create electron-hole pairs, despite the insulating spacers with a thickness of several nanometres [19].

This study reports on charge transfer from a Au plasmon (donor) to a p-$Cu_2O$ semiconductor (acceptor), the mechanisms and benefits of photocatalytic oxidation reactions. Charge transfer was determined by ultrafast spectroscopy and a prevalent mechanism assessed by photocatalysis under different conditions. Since the target reaction is aniline oxidation to azobenzene, holes are needed and thus, to compare reactivity associated with PIRET and hot-carrier injections, one needs to employ a p-type semiconductor since this is the material exposed to the reaction medium. The choice of $Cu_2O$ was motivated by the fact that this was the first material shown to undergo the PIRET process and is also a p-type semiconductor [19]. We do not anticipate any other significant advantage in using p-$Cu_2O$ when it comes to reactivity in aniline oxidation. Copper (I) oxide ($Cu_2O$) exhibits absorption in the visible range, is cheap and abundant, and shows good interaction with $CO_2$ [20,21], making it a great candidate for photocatalysis. Even though this material shows good photocatalytic behaviour, its photo response and stability are challenges that need to be resolved, making it an ideal candidate for this study.

## 2. Materials and Methods

The samples were prepared on fluorine-doped tin oxide (FTO) glass. Before sample preparation, the glass was cleaned following the basic procedure of 30 min sonication in a 2% detergent solution (Hellmanex III, Hellma Analytics) in Milli Q water. This step was followed by cleaning and getting rid of excess detergent with water and an additional 20 min of sonication in Milli Q. The final step was 15 min of sonication in isopropanol and drying in argon gas.

The synthesis Au NPs followed the Turkevisch method [22]. For the reaction 50 mL of trisodium citrate solution ($Na_3C_6H_5O_7$; 6.6 mM) was heated to 70 °C under constant stirring. Once at constant temperature, 0.1 mL of a tannic acid solution ($C_{76}H_{52}O_{46}$; 2.5 mM) was added followed by the immediate addition of 1 mL of chloroauric acid ($HAuCl_4$; 25 mM). After the addition of the chloroauric acid, the solution turned dark blue which shifted to a red colour within the following minutes. The obtained solution was stirred, slowly cooled down to room temperature, and placed in the fridge afterwards to rest for 24 h to ensure a completed reaction.

For the attachment of the Au NPs, the clean FTO glass was subjected to 15 min treatment of UV-Ozone. Subsequently, the Au NPs suspension was mixed with 0.1 M $HNO_3$ in a 5 μL nitric acid ratio for each 100 μL Au NPs suspension. The prepared glass was then immersed in the solution overnight. Finally, the samples were dried in argon and annealed at 450 °C for 30 min.

The $Cu_2O$-samples, used for further investigation, were produced via chemical electrodeposition, using a variation of the procedure published elsewhere [23]. For the deposition, a solution containing 50 mM $CuSO_4$ and 25 mM trisodium citrate was prepared. The precursor solution pH, deposition potential, and time were optimized to get the best possible film in terms of crystallinity and FTO glass coverage. To adjust the precursor solution pH a 1 M NaOH solution was used. Thereby, one needs to take care that the solution stays transparent and does not form any precipitation which would indicate the formation of $Cu(OH)_2$. A three-electrode setup was used to achieve the homogenous semiconductor films, using FTO or FTO/Au-NP films as a working electrode, Ag/AgCl (3.5 M) as a reference electrode, and a film of FTO modified with platinum-NPs as a counter electrode. The deposition area was fixed at 3 $cm^2$. For the deposition of Au-NP on FTO, the deposition potential was reduced to $E = -0.18$ V vs. Ag/AgCl to achieve films of similar quality. Scheme 1 shows the prepared film structure of FTO/Au NPs/p-$Cu_2O$.

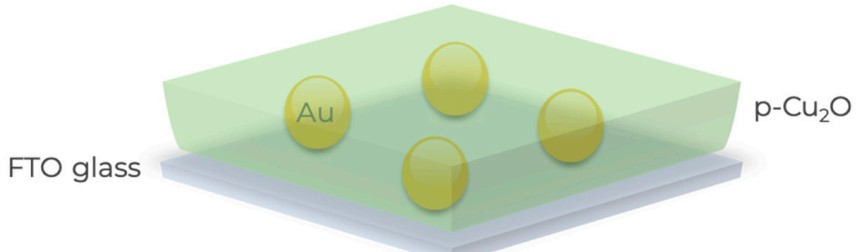

**Scheme 1.** Schematic representation of FTO/Au NPs/p-Cu$_2$O film structure.

The Au NPs were characterized via dynamic light scattering (DLS) from Malvern and by optically using the Ocean Insight UV-Vis spectrometer. The films were characterized optically (UV-Vis Cary 2000 spectrometer) and morphologically via scanning electron microscopy (SEM) on a Zeiss Merlin, SEM Instrument. The crystal structure of the Cu$_2$O phase was determined via grazing incident X-ray diffraction (GIXRD) (Siemens D5000 of the diffractometer, scanning in a range of $2\theta$ = 20–80° using a step of 0.02° and an acquisition time of 4 s/step). AC-impedance was performed to determine the type of Cu$_2$O semiconductor on a Metrohm, Autolab PGSTAT204. For the AC-impedance measurement, TBA-PF$_6$ (0.1 M) in acetonitrile and was used, as was a supporting electrolyte which was degassed in argon. The respective samples were measured in a potential range of $E$ = −0.3 V to 0.2 V vs. Ag/AgNO$_3$ reference electrode, with an amplitude of $\Delta E$ = 10 mV in a frequency range of $f$ = 0.1–10,000 Hz. X-ray photoelectron spectroscopy (XPS) was performed by using an Ulvac-Phi Quantera II instrument with monochromatic Al K$\alpha$ radiation. Measurements were carried out under constant neutralization using low-energy electrons and Ar$^+$ ions because of the catalysts' isolation features [24]. Survey scans and details of selected core levels were obtained from the catalysts' surfaces. The adventitious carbon was chosen as a charge reference with the C 1s peak position at 284.8 eV.

Transient absorption spectroscopy (TAS) was performed to assess charge transfer processes. Briefly, 40 fs pulsed laser with 3 kHz repetition rate was generated through the Libra Ultrafast Amplifier System designed by Coherent. Two optical parametric oscillators (TOPAS-prime, Light Conversion) were used to generate either the excitation beam and/or the probe light in the Mid-IR (3000–10,000 nm). For measurements in the UV-NIR detector, both pump and probe lights were redirected to the Newport MS260i spectrograph with interchangeable gratings. The fundamental laser pulse centered at 795 nm passed through the delay stage (1–2 fs step size) and focused on a Sapphire optical window to generate visible light from 350 to 700 nm. The instrument response function obtained for our system was ca. 95 fs. The kinetic traces were fitted with a sum of convoluted exponentials. The sample was pumped at 620 nm to isolate the excitation to only Au plasmonic, i.e., to avoid excitation of Cu$_2$O.

The photocatalytic activity of the films was assessed on the photo-oxidation of aniline to azobenzene [25]. The reaction was followed via in situ UV/Vis spectroscopy (Ocean Insight UV-Vis spectrometer). Briefly, 3 mL of an ethanol solution of aniline (C = 0.1 mol·L$^{-1}$) was placed in a quartz cuvette. The films were placed inside the cuvette and left in for 10 min in the dark. Afterwards the samples were illuminated by monochromatic CW lasers with various wavelengths ($\lambda$ = 445 nm or $\lambda$ = 532 nm) for 60 min. The procedure was used for FTO/p-Cu$_2$O and FTO/Au NPs/p-Cu$_2$O. Furthermore, as control experiments, the solutions were illuminated without any substrate.

## 3. Results

The optical absorption of the Au NPs is shown in Figure 1. The UV-Vis spectrum is dominated by the LSPR absorption centered at around $\lambda$ = 525 nm and the broad-absorption tail at high energy, related to the interband contribution [26]. DLS analysis revealed that the colloidal Au had a narrow size distribution with its main fraction measuring a diameter of 10 nm. Note that the UV-vis and DLS in several batches of Au NPs were

identical, suggesting good method reproducibility with respect to size distribution and optical absorption.

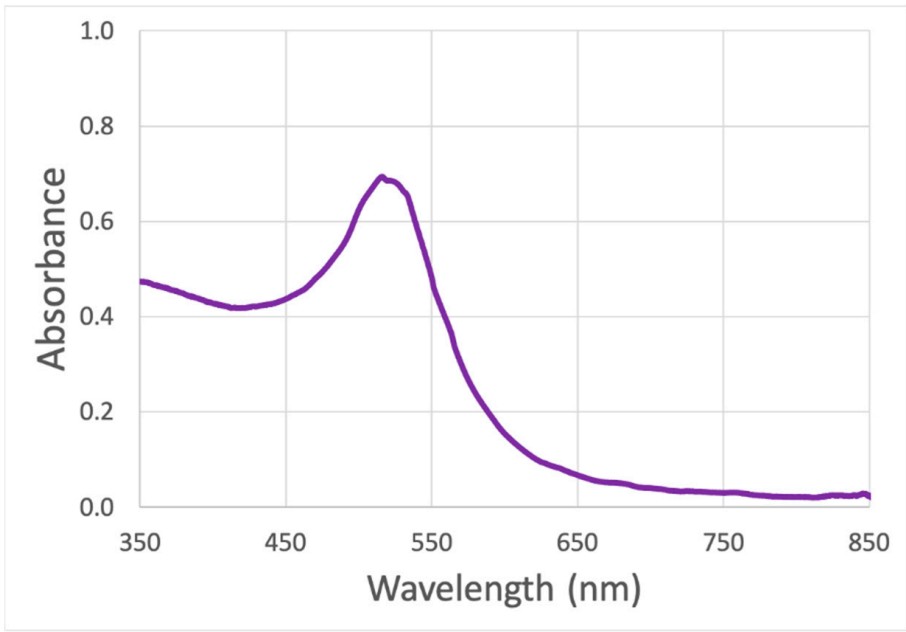

**Figure 1.** UV/Vis-absorption spectrum of the Au NPs' suspension showing its LSPR and inter-band absorptions.

The electrodeposition of $Cu_2O$ films was optimized to prepare pristine p-type $Cu_2O$ cubic structure and ensure good FTO glass coverage. The optimal film quality was achieved with a precursor solution pH of 10, $-0.35$ V vs. Ag/AgCl, and 15 min deposition time. UV-Vis analysis (Figure 2) shows an evident band-edge characteristic of a semiconductor consistent with what was reported elsewhere [23]. The bandgap of the electrodeposited $Cu_2O$ was estimated to be 2.46 eV using a Tauc plot of the data shown in Figure 2.

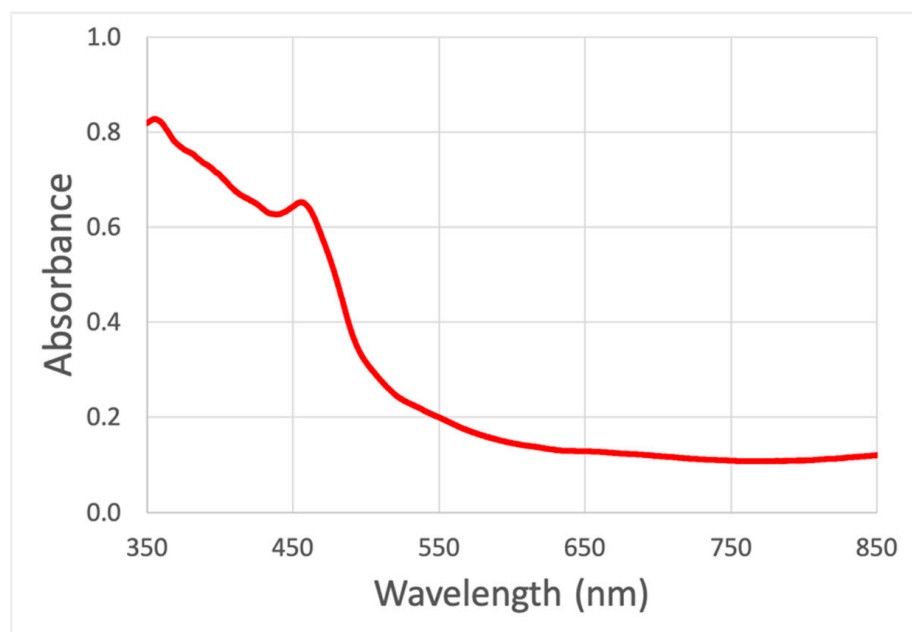

**Figure 2.** UV/Vis-absorption spectrum of $Cu_2O$ film deposited using a precursor solution at pH 10, $-0.35$ V vs. Ag/AgCl, and 15 min deposition time.

SEM assessed the morphology of the deposited $Cu_2O$. It is noticeable from Figure 3 that $Cu_2O$ is deposited in the form of nanocubes with a relatively narrow-edge size distribution of around 30–40 nm. While the material density of FTO is relatively high, there are zones of exposed FTO, as noted in the high magnification Figure 3b. The GIXRD diffractogram presented in Figure 4 established pure cubic $Cu_2O$ with the respective (110, 111, 200, and 311) crystalline planes as the crystal phase [27] resultant from the electrodeposition method. Figure 4 also shows the importance of precursor pH in the final material crystallinity quality since depositions at lower pH (e.g., pH 7) resulted in a material with significantly less intense diffraction peaks.

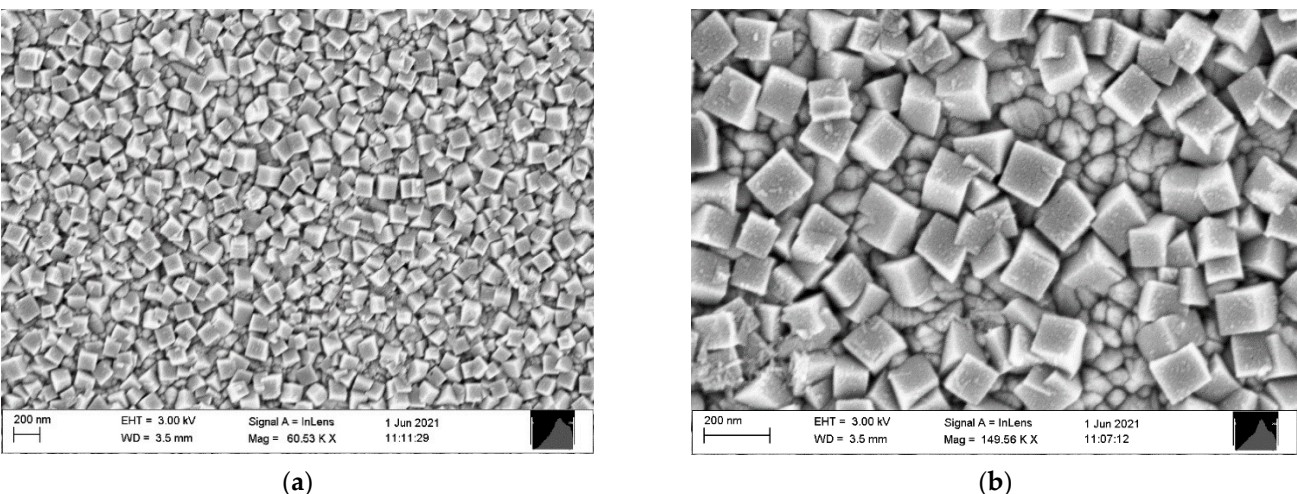

|  |  |
|:--:|:--:|
| (**a**) | (**b**) |

**Figure 3.** SEM image of the $Cu_2O$ substrate after 15 min of deposition at pH 10 and −0.35 V vs. Ag/AgCl in two different magnifications. (**a**) low magnification image (**b**) high magnification image.

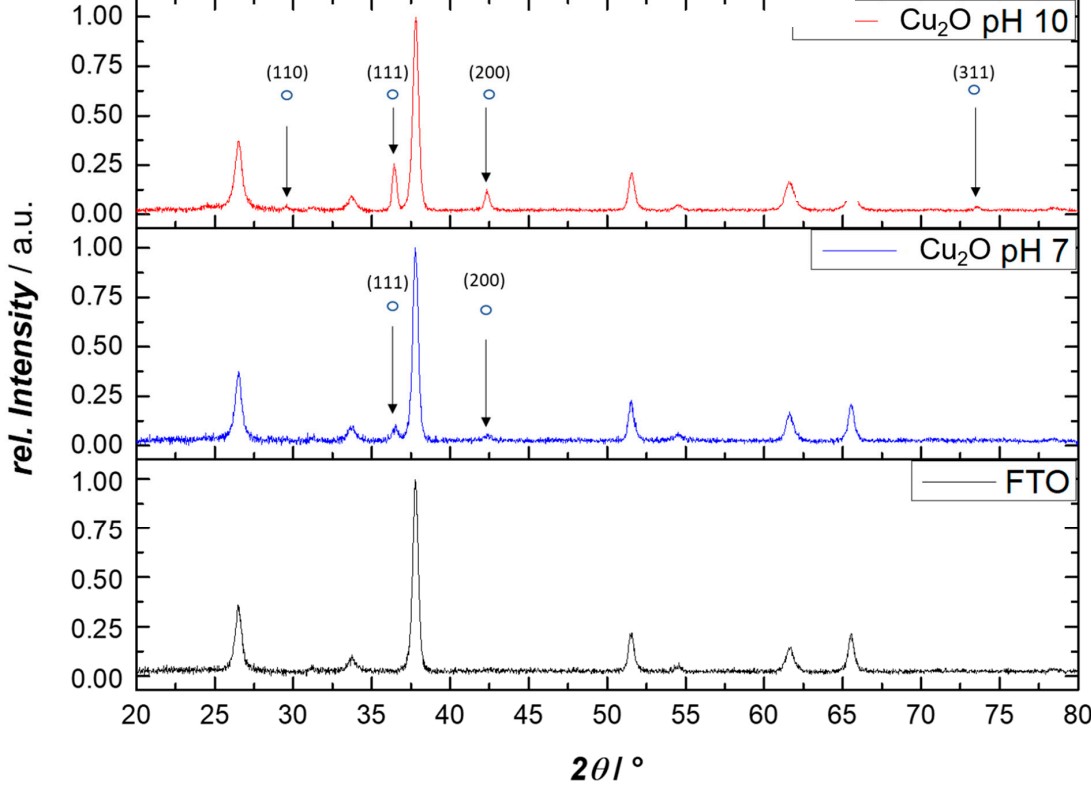

**Figure 4.** GIXRD pattern of the electrodeposited $Cu_2O$-films deposited at pH 7 and pH 10 and the pattern of FTO as reference. Peak assignments were performed based on previously reported literature [27].

According to scientific literature, the environment during electrodeposition allows control of whether one acquires an n-type or a p-type semiconductor [28]. Therefore, it is important to establish what type of $Cu_2O$ was deposited. Taking advantage of AC-impedance measurements and the Mott-Schottky analysis [23,29], the resulting plot gives an indication of which type of semiconductor was deposited. The negative slope with increasing voltage shown in Figure 5 is consistent with p-type conductivity. Note that precursor solution pH does not affect the type of semiconductor deposited.

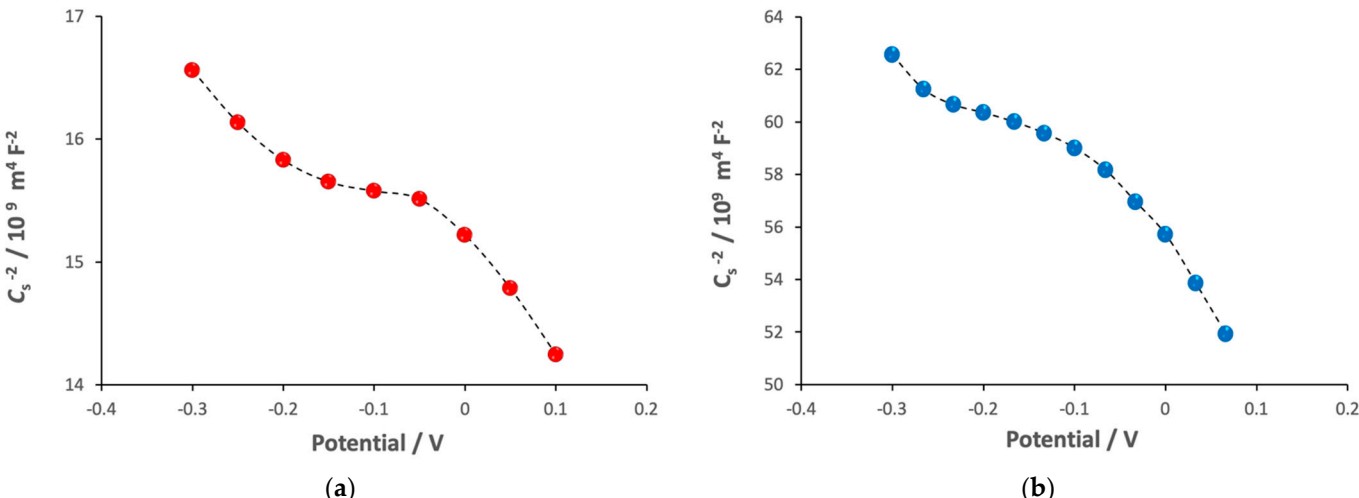

**Figure 5.** Mott–Schottky plot of the electrodeposited $Cu_2O$ films at $-0.35$ V vs. Ag/AgCl for 15 min at (**a**) pH 10 and (**b**) pH 7. Analysis made at 1 kHz for both samples.

A $Cu_2O$ film was deposited on Au NPs supported on FTO, upon adjusting the deposition pH. Figure 6 shows the optical properties of the deposited film. The deposit film shows the characteristic $Cu_2O$ absorption band edge which corroborates its deposition. As mentioned previously, the plasmonic resonance peak position depends on the dielectric media, which become apparent when embedding the Au NPs in the semiconductor. In solution or on FTO film, the Au NPs are mostly surrounded by water from a solution or air humidity, with a dielectric constant of $\varepsilon_r \approx 78$ [30]. When $Cu_2O$ is deposited on top of Au NPs, the NPs' dielectric media change to $\varepsilon_r \approx 7.5$ [31] causing a significant redshift in Au plasmon absorption (red trace) [32]. The shift is not related to the damage or sintering of Au NPs because the absorption of the Au NPs reverts to its original value after the chemical etching of $Cu_2O$ with an ammonia solution (light blue trace).

The UV-Vis analysis suggested that $Cu_2O$ is deposited on top of Au NPs, resulting in a complete immersion of Au NPs. An XPS analysis was performed to confirm this. Figure 7 shows the Cu 2p and Au 4f XPS detail scans for the FTO/Au NPs/p-$Cu_2O$ and FTO/Au NPs after the etching of p-$Cu_2O$. There are prominent peaks associated with Cu 2p in the FTO/Au NPs/p-$Cu_2O$ film (Figure 7a). The Cu $2p_{3/2}$ peak centred at 932.3 eV binding energy ascribed to the $Cu^+$ species from $Cu_2O$ [33], is corroborated by the satellite feature between 940–945 eV that is absent in metallic copper. As expected, the Cu 2p signal almost disappears after the chemical etching. A nearly opposite situation happens with Au 4f signal (Figure 7b). In other words, in the FTO/Au NPs/p-$Cu_2O$ there is no evidence for Au, while its doublet is easily seen after the chemical etching of p-$Cu_2O$. Note that XPS has a probing depth of ca. 10 nm and since the $Cu_2O$ is present in the form of nanocubes with edges around 30–40 nm, the only way that Au could be visible is if $Cu_2O$ did not cover the complete surfaces of the Au NPs. This observation corroborates the UV-Vis finding that the electrodeposited $Cu_2O$ completely covers Au NPs. The Au $4f_{7/2}$ on the etched sample has a binding energy of 84.1 eV consistent with metallic gold [34].

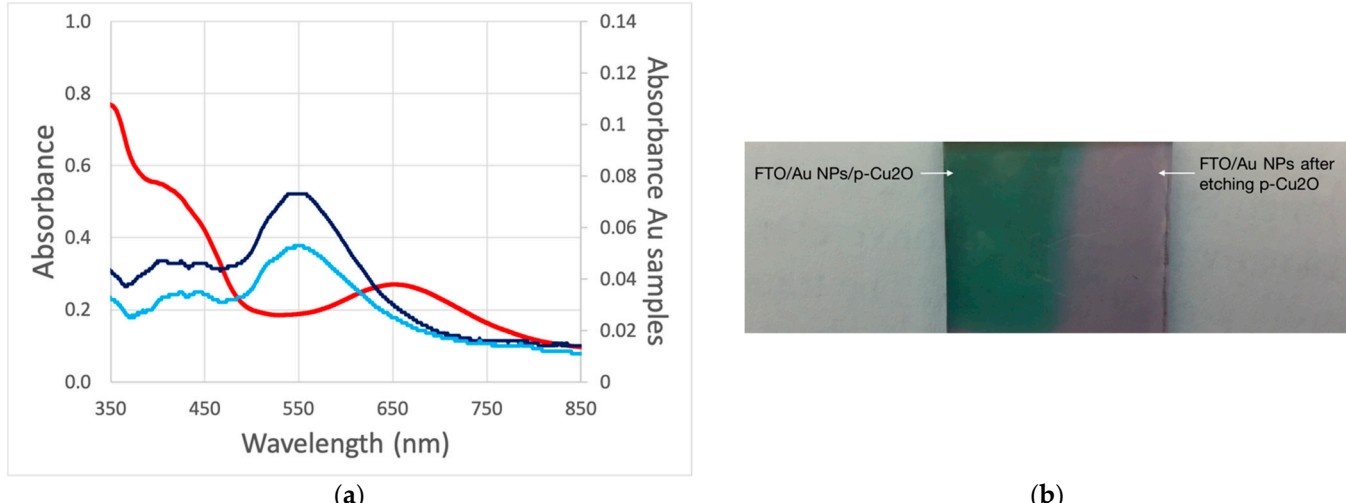

**Figure 6.** Optical properties of FTO/Au NPs/p-Cu$_2$O film: (**a**) UV/Vis of the Au covered in Cu$_2$O (red; primary axis); FTO/Au NPs fresh (dark blue; secondary axis); and FTO/Au NPs after etching with ammonia solution (light blue; secondary axis). (**b**) Photograph of FTO/Au NPs/p-Cu$_2$O film with an edge etched with ammonia.

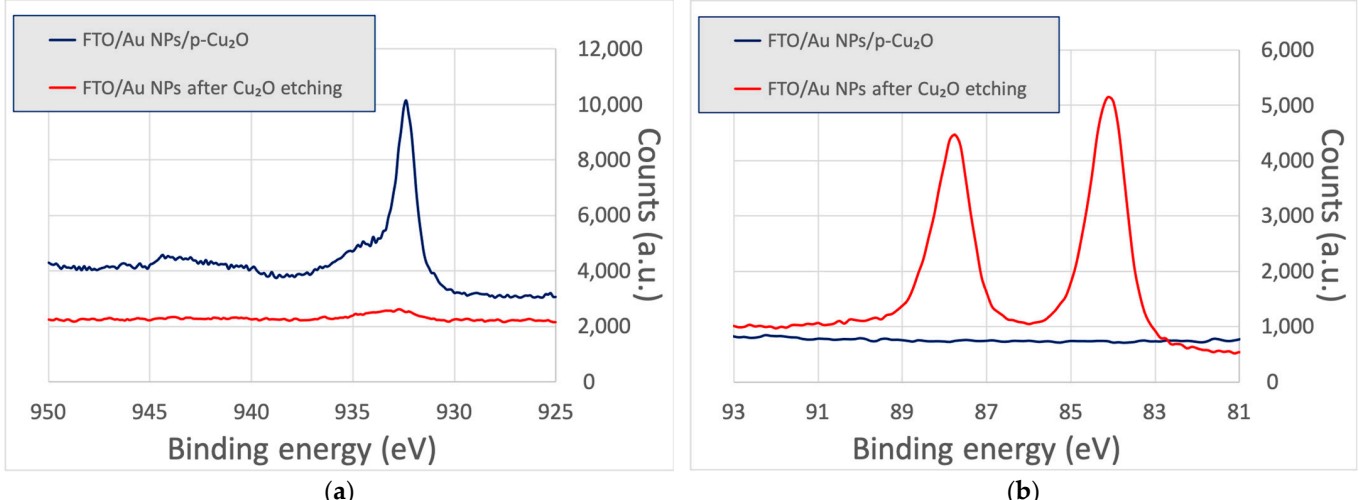

**Figure 7.** XPS detail scan analysis of the FTO/Au NPs/p-Cu$_2$O and FTO/Au NPs after etching pCu$_2$O films: (**a**) Cu 2p$_{3/2}$ region; (**b**) Au 4f region.

Curiously, based on XPS analysis, the electrodeposition of Cu$_2$O seems to cover the FTO throughout since there is no evidence of the characteristic Sn 3d doublet from the tin in the FTO glass (Figure 8), which seems to somehow contradict the SEM images. SEM provides a very local analysis, while the XPS provides an overview of the entire sample, signifying that there might be some pinholes in the film as detected by SEM, but they are not significant when considering the whole sample. As expected, the Sn 3d peaks are clearly visible on the FTO/Au NPs since the Au NPs are sparsely distributed through the film.

Ultrafast transient absorption spectroscopy (TAS) was performed to understand the processes that led to reactive charges on Cu$_2$O originating from Au plasmon excitation. Plasmon excitation resulted in a negative signal (bleach) centred at the laser wavelength excitation and two positive winglets, as reported elsewhere [35–38]. Kinetic traces extracted at the maximum of the winglets enabled us to estimate the electron-phonon (e-ph) lifetime that relates to Au NPs' electronic heat capacity. This is proportional to both the electronic temperature and the electron density, i.e., is highly sensitive to the electronic structure of the metal [39]. Figure 9 shows the kinetic traces extracted at the maximum of the winglet

to the blue of the bleach signal for the FTO/Au NPs/p-Cu$_2$O (orange trace) and FTO/Au NPs (blue trace). The addition of p-Cu$_2$O resulted in a significant decrease in the e-ph lifetime suggestive of hot-hole injection into the semiconductor [11]. This was confirmed by a negative signal centred at 350 nm (Figure 10) associated with a bleached bandgap attributed to a charge injection into the semiconductor [40], which is absent when Au is not present. On the basis of band alignment and that Cu$_2$O is p-type, the injected charge is more likely to be due to hot holes. Any hot electron that is injected is expected to rapidly recombine due to band bending [41]. Additionally, a second bleach signal was observed after 5 ps, which can be assigned to the charge creation of p-Cu$_2$O, i.e., not originated from the hot-carrier injection. This can be ascribed to the PIRET process. The excited carriers of Cu$_2$O have lifetimes above 1 ns, which makes them available for photocatalysis reactions.

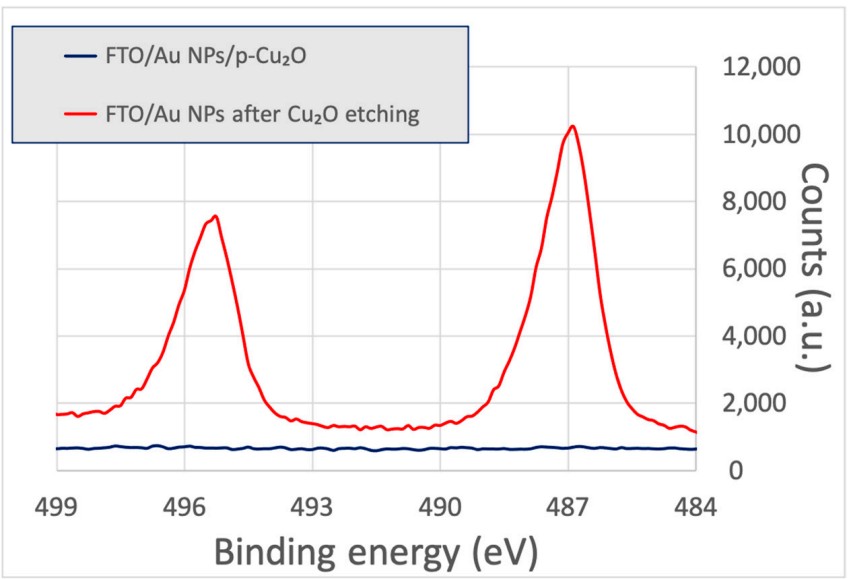

**Figure 8.** Sn 3d XPS detail scan analysis of the FTO/Au NPs/p-Cu$_2$O and FTO/Au NPs after etching p-Cu$_2$O films.

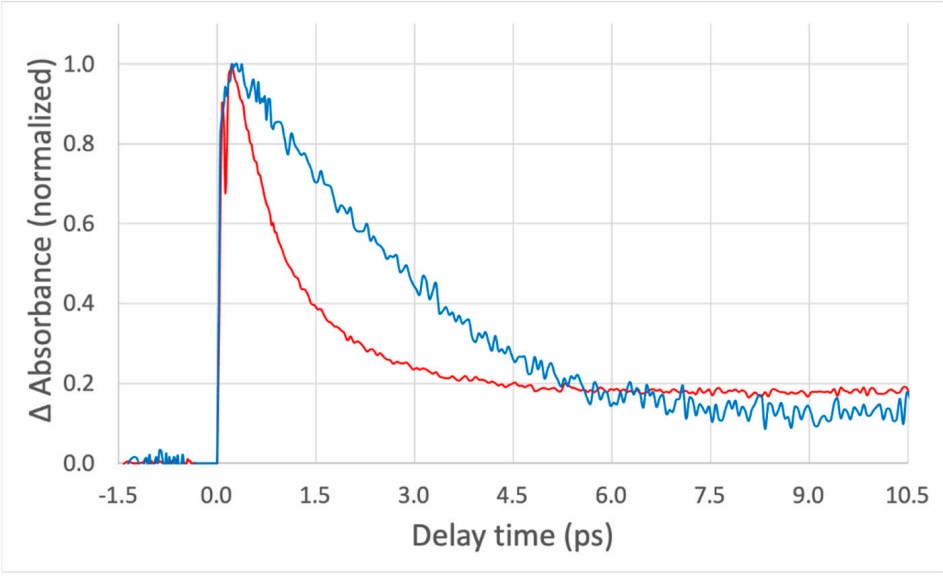

**Figure 9.** Kinetic trace extract at the maximum of the winglet to the blue of the Au plasmon bleach after excitation at 620 nm: FTO/Au NPs (blue trace) and FTO/Au NPs/p-Cu$_2$O (red trace).

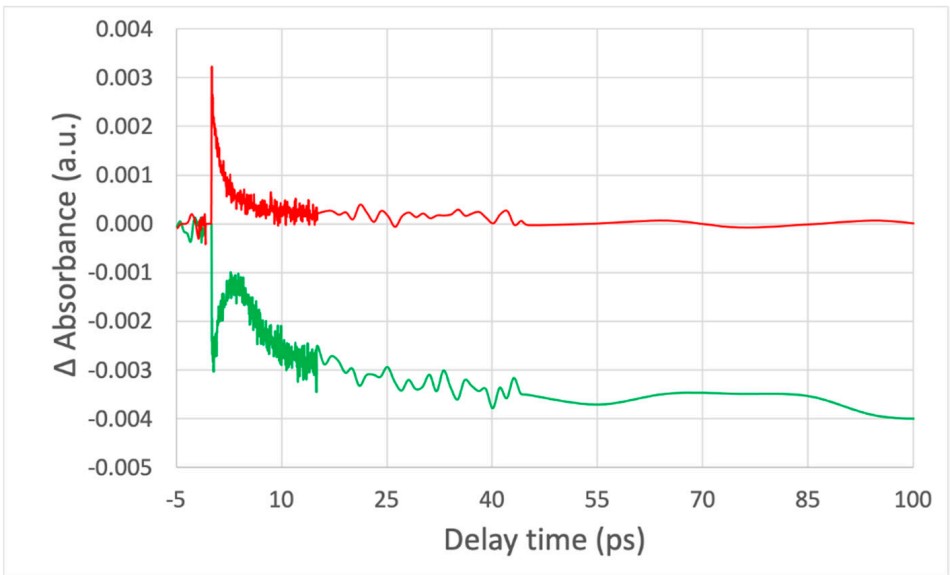

**Figure 10.** Kinetic trace extract at 350 nm after excitation of Au plasmon at 620 nm for FTO/Au NPs/p-Cu$_2$O (green trace). For comparison purposes the signal is overlapped with the trace extracted at the winglet for FTO/Au NPs/p-Cu$_2$O (red trace).

Photo-oxidation of aniline to azobenzene was used for the reactivity of p-Cu$_2$O carriers originated from plasmon excitation. The photoreactivity of p-Cu$_2$O was accessed by exciting the films at 445 nm (bandgap excitation). Figure 11 shows the formation of azobenzene as a function of illumination time at 445 nm. There is an apparent increase in the rate of product formation which is enhanced when Au is present. Note that at 445 nm only semiconductor bandgap excitation is expected to yield photo-active carriers. The observation confirmed that p-Cu$_2$O is active and selective to azobenzene formation. Excitation of the same samples at the Au plasmon resonance (532 nm) resulted in product formation only when Au NPs were present (Figure 12a). The detected signal for bare p-Cu$_2$O was lower than the signal without laser illumination, suggestive of no reaction of the p-Cu$_2$O sample under 532 nm illumination. The signal without laser excitation (Figure 12b) seemed to be related to product formation due to interaction with the UV/Vis probe light. Note that when experiments were performed under 650 nm laser excitation, no product formation was detected.

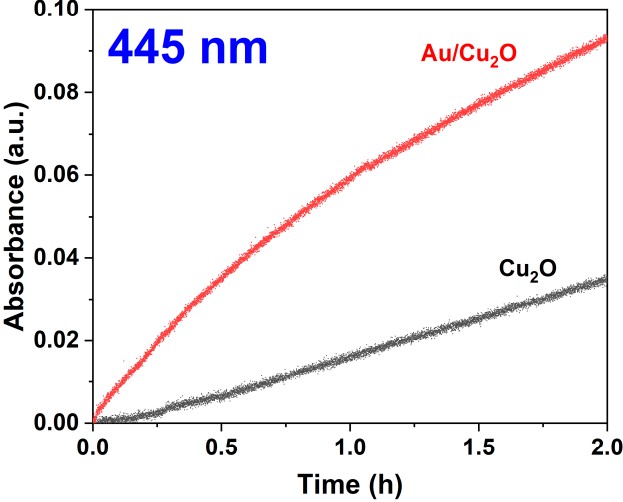

**Figure 11.** Azobenzene formation from aniline photooxidation under 445 nm laser excitation.

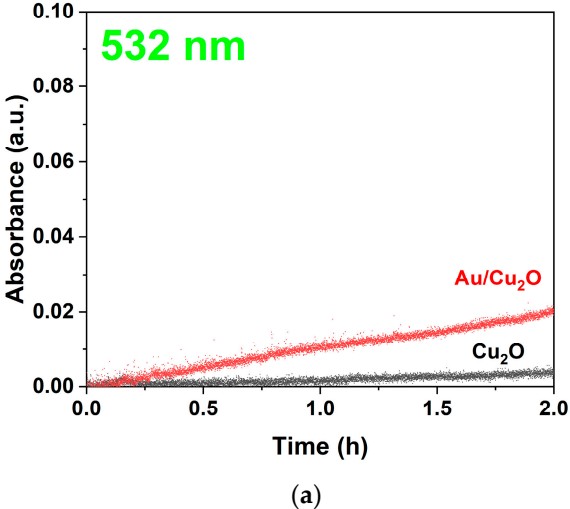
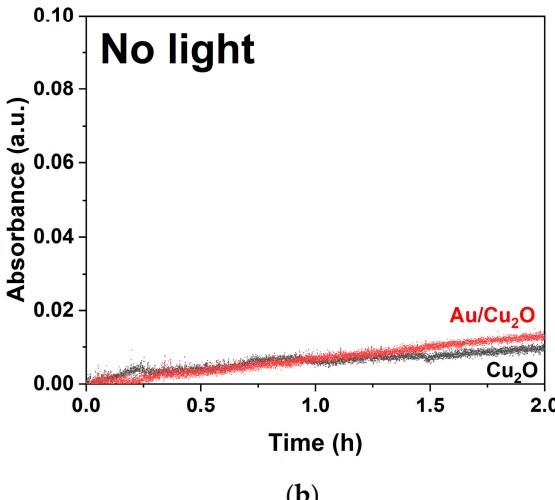

**Figure 12.** Azobenzene formation from aniline photo-oxidation: (**a**) under 532 nm laser excitation; (**b**) without laser excitation.

## 4. Discussion

The detection of azobenzene originates from aniline photo-oxidation when Au/p-$Cu_2O$ is excited at 532 nm, confirming that reactive holes are present on p-$Cu_2O$ from plasmon excitation, which are otherwise not present. According to the ultrafast TAS measurements, the origin of those holes can be from hot-hole injection and the PIRET process. From bandgap excitation photocatalytic reactions in the semiconductor, one can deduce that, when generated in the same materials, electrons and holes recombine readily, reducing the available population of holes that can participate in the oxidation. This is corroborated by the significant increase in reaction rate under 445 nm of illumination when Au is present. In this case, Au NPs act as an electron sink [42], increasing charge separation and, thus, the population of holes available for reaction. Therefore, it is reasonable to assume that the number of holes available for the reaction, which originate from the PIRET process, is relatively small compared with the number of holes generated from the hot-carrier injection. This is because, in the case of PIRET, those holes are generated directly at the semiconductor together with electrons which recombine rather quickly.

Presently, we do not have a concrete answer for the faith of the electrons transferred to Au. However, when the reactions were run for a longer time, a significant decrease in reaction rate was observed. This suggests that after reaching a certain threshold, the electrons in Au started competing for holes, i.e., foment interfacial charge recombination. However, one cannot discard the possibility that this decrease is due to film instability.

The absence of product when the experiments were performed under 650 nm laser excitation was partly justified by the experimental setup because the films were illuminated from the FTO glass side of the film (FTO/Au/p-$Cu_2O$). A similar result was observed for methylene-blue photo bleach, suggesting that effective LSPR excitation occured when exciting Au NPs on its original absorption.

The final aspect relates to the possible involvement of heat generated from plasmon excitation in the reaction. Non-radioactive decay of LSPR results in heat generation can promote catalysis. We do not discard this possibility, but to produce azobenzene holes must be involved. The focus of this work is to rationalize the origin of the holes involved in the process, not to determine actual rates of reaction. This is a highly complex endeavor when talking about plasmon-driven reactions. We suspect that heat would promote the reaction equally if the holes came from PIRET or hot-hole injection, since it would be on molecular and solvent dynamics, not on the carrier formation.

## 5. Conclusions

To summarize, the addition of Au to p-Cu$_2$O expanded the semiconductor light absorption range and photocatalytic window. The excitation of Au plasmon on the Au/p-Cu2O system resulted in hot-hole injection and electron-hole pairs coming from the PIRET process. The holes were found to be active in the aniline oxidation of azobenzene. However, based on processes dynamics, the large amount of active charge utilized in the reaction originates from hot-hole injection, not the PIRET mechanism. This study showed that to utilize the charge solely generated from PIRET process, the plasmonic material must be covered by an insulating layer, as proposed elsewhere [19]. Additionally, the most significant benefit of the PIRET process is that photocatalytic reactions, which require both electrons and holes, obtain the final product.

**Author Contributions:** Conceptualization, P.H. and J.S.; methodology, P.H., V.R.R.S., R.B.V. and E.L.; validation, P.H., V.R.R.S., R.B.V. and J.S.; formal analysis, P.H., V.R.R.S., R.B.V. and J.S.; investigation, P.H., V.R.R.S., R.B.V. and E.L.; resources, J.S.; data curation, P.H. and J.S.; writing—original draft preparation, J.S.; writing—review and editing, all authors; visualization, P.H., V.R.R.S., R.B.V. and J.S.; supervision, J.S.; project administration, J.S.; funding acquisition, J.S. All authors have read and agreed to the published version of the manuscript.

**Funding:** This research was funded by Uppsala University internal funds, Knut & Alice Wallenberg foundation (grant no. KAW 2019.0071) and Swedish Research Council (grant no. VR 2019-03597).

**Institutional Review Board Statement:** Not applicable.

**Informed Consent Statement:** Not applicable.

**Acknowledgments:** The authors would like to thank Uppsala University, KAW, and VR for their support.

**Conflicts of Interest:** The authors declare no conflict of interest.

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
