# Peer review of "Plasmon-Mediated Oxidation Reaction on Au/p-Cu2O: The Origin of Hot Holes"

_2673-7167, doi:10.3390/physchem1020011_

Round 1

Reviewer 1 Report

Plasmon induced hot holes accumulation has been demostrated in FTO/Au NPs/p-Cu2O system. These hot holes used for photocatalytic oxidation of aniline to azobenzene. I recommend this article for its acceptance in Physchem after minor revision:

1) Author told that Au nanocrystals act as electron sink. Hot holes generated in plasmonic excitation of Au are trasfered to p-Cu2O then excess of hot holes present on Cu2O surface carried out photooxidation reaction. If it is so then what is the fate of hot electrons accumulated in Au nanocrystals?

2) In FTO/Au NPs/p-Cu2O, plasmonic peak for Au nanocrystals (525 nm) redshifted to 655 nm upon formation of Cu2O shell. Then why the photooxidation of aniline did not carried out using 655 nm laser? As it is the λmax for FTO/Au NPs/p-Cu2O, therefore at this wavelength maximum photooxidation reaction rate would be observed.

3) What is the thickness limit for Cu2O shell on Au nanocrystals to see the light induced hot hole transfer process?

Author Response

Please find reply to comments in the attachment. 

Reviewer 2 Report

The manuscript with ID “Physchem-1312514” and entitled “Plasmon mediated oxidation reaction on Au/p-Cu2O: the origin of hot holes” is an interesting work. I only have some suggestions to improve this report before publishing it in the journal “Nanomaterials.”

  1. The introduction has a lengthy discussion of the mechanism of the plasmonic photocatalytic reaction. However, it is the too-short paragraph of why using p-Cu2O and Au decoration in this research. The authors are suggested to balance this. It is also suggested to talk about using the Au covering by p-Cu2O but not Au on p-Cu2O.
  2. Why used p-type Cu2O? What is the advantage of using p-type Cu2O for the plasmonic photocatalytic reaction?
  3. In line 214, “… -0.35 vs Ag/AgCl ...” The writing is ambiguous.
  4. In Fig. 4, the authors presented sample XRD fabricated by pH 7 and 10 precursors. Were they fabricated on the FTO substrate with Au NPs? What was the difference in photocatalytic processing efficiency?
  5. A legend is needed in Fig. 6(a)
  6. In line 334, the “Ultrafast transient absorption spectroscopy (TAS) was performed to understand the processes that lead to reactive charges on Cu2O originated from Au plasmon excitation.” In line 337, “Kinetic traces extracted at the maximum of the winglets enable us to estimate electron-phonon (e-ph) lifetime that relates to Au NPs electronic heat capacity, ….” The authors should provide the “ultrafast transient absorption spectroscopy” of the two samples in Figure 9 as the “maximum of the winglets” in the spectroscopy data were used. The reference works are not the authors’ previous works.
  7. In the caption of Figure 10, the sample are all “FTO/Au NPs/p-Cu2O”?
  8. The absorption spectrum did not present the difference of adding Au NPs to the p-Cu2O layer on FTO.
  9. Following the #8 comment. The absorption spectrum of the various samples did not suggest that the 445 nm light-triggered plasmonic enhancement of the photooxidation of aniline to azobenzene in line 358. The Au/Cu2O in Figure 12 (b) presented a “dark” catalytic oxidation of aniline to azobenzene. It can also say that the 532 nm light illuminating on Au NPs converted to heat and raise the temperature of the sample. The temperature rises enhanced the catalytic oxidation of aniline to azobenzene. No from the light generated hot charges on Au triggers the plasmonic photocatalytic reaction. It might not be PIRET claimed in line 381.
  10. Why the catalytic oxidation of aniline to azobenzene in “dark” in Figure 12 (b) presented somehow higher efficiency than that “under light illumination” in Figure 12 (a).

Author Response

(The authors gave the same response as above.)

Round 2

Reviewer 2 Report

I have no more questios.